# Breast Cancer Survivors and Healthy Women: Could Gut Microbiota Make a Difference?—“BiotaCancerSurvivors”: A Case-Control Study

**DOI:** 10.3390/cancers15030594

**Published:** 2023-01-18

**Authors:** Telma Caleça, Pedro Ribeiro, Marina Vitorino, Maria Menezes, Mafalda Sampaio-Alves, Ana Duarte Mendes, Rodrigo Vicente, Ida Negreiros, Ana Faria, Diogo Alpuim Costa

**Affiliations:** 1Medical Oncology Department, Hospital Professor Doutor Fernando Fonseca, 2720-276 Amadora, Portugal; 2Laboratory Medicine Centre Germano de Sousa, 1600-513 Lisbon, Portugal; 3Medical Oncology Department, Hospital do Espírito Santo de Évora, 7000-811 Évora, Portugal; 4PTSurg–Portuguese Surgical Research Collaborative, 1600 Lisbon, Portugal; 5Faculdade de Medicina da Universidade do Porto, 4200-319 Porto, Portugal; 6Hospital CUF, Breast Cancer Unit, 1998-018 Lisbon, Portugal; 7Faculdade de Ciências Médicas, NOVA Medical School, 1169-056 Lisbon, Portugal; 8Comprehensive Health Research Centre (CHRC), Faculdade de Ciências Médicas, NOVA Medical School, 1150-082 Lisbon, Portugal; 9CINTESIS-Center for Health Technology Services Research, Faculdade de Ciências Médicas, NOVA Medical School, Universidade NOVA de Lisboa, 1169-056 Lisboa, Portugal; 10Medical Oncology Department, Hospital de Cascais Dr. José de Almeida, 2755-009 Cascais, Portugal

**Keywords:** microbiota, microbiome, breast cancer, cancer, case-control study, survivors, survivorship

## Abstract

**Simple Summary:**

Breast cancer (BC) is the most commonly diagnosed cancer and the second cause of cancer-specific death in women worldwide. Increasing evidence suggests that gut microbial dysbiosis may have a role to play in the pathogenesis, treatment, and prognosis of BC. The “BiotaCancerSurvivors” was a prospective, longitudinal, observational, unicentric, and case-control study that aimed to analyse whether the gut microbiota differs between cancer survivors and a database of healthy controls.

**Abstract:**

In this first analysis, samples from 23 BC survivors (group 1) and 291 healthy female controls (group 2) were characterised through the V3 and V4 regions that encode the “16S rRNA” gene of each bacteria. The samples were sequenced by next-generation sequencing (NGS), and the taxonomy was identified by resorting to Kraken2 and improved with Bracken, using a curated database called ‘GutHealth_DB’. The α and β-diversity analyses were used to determine the richness and evenness of the gut microbiota. A non-parametric Mann-Whitney *U* test was applied to assess differential abundance between both groups. The Firmicutes/Bacteroidetes (F/B) ratio was calculated using a Kruskal-Wallis chi-squared test. The α-diversity was significantly higher in group 1 (*p* = 0.28 × 10^−12^ for the Chao index and *p* = 1.64 × 10^−12^ for the ACE index). The Shannon index, a marker of richness and evenness, was not statistically different between the two groups (*p* = 0.72). The microbiota composition was different between the two groups: a null hypothesis was rejected for PERMANOVA (*p* = 9.99 × 10^−5^) and Anosim (*p* = 0.04) and was not rejected for β-dispersion (*p* = 0.158), using Unifrac weighted distance. The relative abundance of 14 phyla, 29 classes, 25 orders, 64 families, 116 genera, and 74 species differed significantly between both groups. The F/B ratio was significantly lower in group 1 than in group 2, *p* < 0.001. Our study allowed us to observe significant taxonomic disparities in the two groups by testing the differences between BC survivors and healthy controls. Additional studies are needed to clarify the involved mechanisms and explore the relationship between microbiota and BC survivorship.

## 1. Introduction

Female breast cancer (BC) has now surpassed lung cancer as the most commonly diagnosed malignancy, with 2.3 million new cases worldwide. Furthermore, it is the second leading cause of cancer-related death among women, accounting for 685,000 deaths [1]. Access to healthcare systems, population ageing, and improvements in early detection and treatment personalisation have led to a growing prevalence of cancer survivors.

Faced with the new challenges imposed by this trend, since the 1980′s, the survivorship definition has emerged [2] and evolved over time, and several definitions can be found in the literature [3]. The most consensual describes the survivorship experience as a period focused on the health and well-being of a person with cancer from diagnosis until the end of life. It also integrates cancers’ physical, mental, emotional, social, and financial effects that begin at diagnosis and continue through treatment and beyond [3].

In fact, BC survivors make up the largest population in the cancer survivor community. Current data show that up to 85.5% of women with BC will survive 5 years and up to 70% 10 years after diagnosis [4]. Unfortunately, due to the BC trajectory, these patients often experience decrements in quality of life [5,6,7,8]. Some reports have mentioned an improvement in quality of life after cessation of treatments. However, concerns about this issue persist long after treatment, including psychological and physical care.

Genetic and other established risk factors, such as a high body mass index (BMI) and a sedentary lifestyle, have been associated with the onset and progression of BC. Diet and exercise have been demonstrated to affect disease-free survival in BC. In addition to these well-established risk factors, accumulating studies have indicated that local and gut alterations may contribute to the pathogenesis of both gastrointestinal (GI) and extra-intestinal tumours, such as BC [9].

In recent years, the role of the gut microbiota and its relation to BC has become a significant area of interest in medical research [10,11].

The microbiota refers to the microbes resident on and inside the body. Different microbiota ecosystems exist, such as the GI tract, skin, or oral cavity, estimating trillions of microorganisms among various locations in the human body [12,13,14]. The gut microbiota is unique in each individual and is determined by lifestyle and genetic factors, such as physical activity or maternal microbial composition [15]. This high interindividual variation in the healthy population makes defining dysbiosis a challenge, with no clear definition of the healthy gut microbiota yet established. Microbial dysbiosis results when maladaptation or abnormal composition occurs within the microbial community of a given organ or tissue and has recently been considered a critical factor in cancer development. It has been shown that the constitution of the microbiota influences tumour biology, such as oncogenic signalling, drug metabolism, or immune system regulation. All these aspects will contribute to the cancer development, growth, and treatment response [16].

For microbiota analysis, it is important to know the principal concepts that allow us to understand the differences between a homeostatic and a dysbiotic microbiota (α and β-diversity). While α-diversity is a measure of microbiota diversity (number of different species in the gut) calculated by many indexes, β-diversity measures the similarity of two communities being used to compare samples [11]. A pilot study in postmenopausal women showed that those with BC had a statistically significant modified composition (β-diversity) and lower oestrogen-independent α-diversity, which means less microbial richness and diversity [17].

As mentioned, gut microbiota could modulate BC risk since it has an important role in the metabolism and secretion of hormone-like bioactive compounds, such as reactivated oestrogens, active phytoestrogens, lithocholic acid, cadaverine, and short-chain fatty acids (SCFAs) [10,18].

Breast microbiota could also influence carcinogenesis by enhancing the local exposure of breast tissue to a hormonal trigger. Oestrogen metabolism–gut microbiota dysfunction combined with individual variations in oestrogen levels may contribute to an increased risk of BC. On the other hand, specific types of intestinal bacteria metabolise phytoestrogens (e.g., isoflavones) and convert them into active metabolites that can protect against BC [19].

The microbiota does indeed have a well-established role in tumour carcinogenesis. Moreover, due to the interference with therapeutics, which can change therapeutic effects, microbiota should be considered a possible ally in treating cancer patients.

Therefore, we hypothesised that the gut microbiota differed between patients with BC survivors and healthy women. To test this hypothesis, we characterised and compared the microbiota in stool samples from BC survivors and faeces from control women using 16S rRNA gene sequencing.

## 2. Materials and Methods

The BiotaCancerSurvivors was a prospective, longitudinal, observational, unicentric and case-control study that aimed to analyse whether the gut microbiota differed between cancer survivors and a database of healthy controls. In this first analysis, samples from 23 BC survivors (group 1) treated at Cuf Oncologia and 291 healthy female controls (group 2) were included.

### 2.1. Patient Enrollment

Enrollment was open to voluntary women aged over 18 with a BC diagnosis who had completed their core treatments (surgery, chemotherapy, and/or radiotherapy). Endocrine therapy (ET) at the moment of sample collection was allowed. Patients were excluded if they were under the age of 18, had a history of other cancers, were pregnant, had received pre- or post-biotic medication within 6 months, or were non-compliant.

### 2.2. Patient Data

Patient data, namely demographics, comorbidities, chronic and recent medication, social habits, menopausal status, type of delivery and duration of lactation, anthropometric measurements, and family health history, were obtained from a written questionnaire. The date of diagnosis, clinical and pathological staging, tumour histological, and molecular characteristics (grade, diameter, lymph node involvement, Ki-67 index, oestrogen receptor (ER)/progesterone receptor (PR) expression, and human epidermal growth factor receptor-2 (HER2) status, were obtained from the consultation of patients’ electronic records, including imaging tests and pathology reports.

### 2.3. Control Group

A database of 291 healthy female samples, selected from the American Gut Project, was used as a control population, filtered to present healthy values of relative abundance in concordance with its BMI and the absence of pathologies, such as small intestinal bacterial overgrowth, irritable bowel syndrome, inflammatory bowel disease, diabetes, and autoimmune diseases.

### 2.4. Sample Collection, DNA Extraction, and 16S rRNA Gene Sequencing and Analysis

Regarding the control group, all samples were processed using the Earth Microbiome Project (EMP) protocols. Briefly, the V4 region of the 16S rRNA gene was amplified with barcoded primers and initially sequenced using the 515f/806r primer pair with the barcode on the reverse primer, and subsequent rounds were sequenced with the updated 515f/806rB primer pair with the barcode on the forward read. Samples were sequenced using Illumina technology [20].

The 16S sequence data were processed using a sequence variant method, Deblur (v1.0.2, Knight Lab, Boulder, CO, USA), trimming to 125 nucleotides (nt), to maximise the specificity of 16S data; a trim of 125 nt was used because one sequencing round in the American Gut used 125 cycles while the rest used 150. Following processing by Deblur, previously recognised bloom sequences were removed. The Deblur sub-operational taxonomic units were inserted into the Greengenes 13_8 99% reference tree using the SATé-enabled phylogenetic placement (SEPP). The SEPP uses the simultaneous alignment and tree estimation strategy to identify reasonable placements for sequence fragments within an existing phylogeny and alignment. Taxonomy was assigned using an implementation of the RDP classifier as implemented in QIIME2 [20].

Fresh stool samples without undigested food residues or other solid substances were collected from the 23 survivors who met the inclusion criteria. Samples were stored at ambient temperature, using a stabilisation solution for nucleic acids in biological samples, which preserves genetic integrity and expression profiles at ambient temperatures and completely inactivates infectious agents, and prevents degradation from freeze-thaw cycling and unexpected freezer failures.

DNA extraction was performed using NucliSENS easyMAG, an automated platform for the isolation (purification and concentration) of total nucleic acids (RNA/DNA) from biological specimens, and BOOM^®^ technology, based on the nucleic acid binding property of magnetic silica particles under high salt conditions.

The studied samples were characterised through the V3 and V4 regions that encode the “16S rRNA” gene of each bacteria, and its taxonomy was identified through a database “16S rRNA” + “GreenGenes 13_8” named GutHealth_DB.

The sample was sequenced by next-generation sequencing (NGS), using the IonTorrent S5 platform, and amplified with the Ion 16S™ Metagenomics kit. The taxonomy was identified by resorting to Kraken2 and improved with Bracken, resorting to the classification database ‘GutHealth_DB’.

### 2.5. Alpha and Beta Diversity Analysis

To quantify the α-diversity richness and evenness of gut microbiota, three different indices (Chao1, Shannon, and ACE) were used.

Aiming to compare the similarity between samples, β-diversity was investigated using Unifrac weighted distance, which in our study was displayed in the form of principal coordinate analysis (PCoA). The approaches used to detect differences in β-diversity between groups and obtain *p*-values were permutational multivariate analysis of variance (PERMANOVA) and similarities of the analysis of similarities (Anosim).

### 2.6. Statistical Analysis

Regarding patient data, IBM SPSS Statistics, (v28.0.1.0, IBM, Armonk, NY, USA), was used to analyse the demographic and tumour characteristics.

Given the sample size, data analysis using a non-parametric Mann-Whitney *U* test was performed to assess the differential abundance between both groups. The Firmicutes/Bacteroidetes (F/B) ratio was calculated using a Kruskal-Wallis chi-squared test. A *p*-value ≤ 0.05 was considered statistically significant. The figures were prepared using R (v3.2.0, www.r-project.org, (accessed on 12 February 2022)). To display graphical representation, ggplot2 was used.

### 2.7. Research Ethics

The study was conducted in accordance with the fundamental ethical principles for medical research involving human subjects, as stated in the Declaration of Helsinki (World Medical Association-2013). It was approved by the Ethics Committee of CUF Descobertas Hospital in Lisbon, Portugal. Furthermore, written informed consent was obtained from all the participants.

## 3. Results

### 3.1. Control Group Characteristics

The control population consisted of 291 healthy females, with a median age of 44 years (range 18–71) and a median BMI of 21.31 (range 19.04–24.55).

### 3.2. Patient Characteristics

Twenty-three female BC survivors were enrolled in this study, with a median age of 53 years (range 38–76). The median BMI was 26.10, with a minimum value of 21.60 and a maximum value of 41.60. In the study population, 47.3% (*n* = 11) of patients were within healthy weight (18.5–24.9), and 34.80% (*n* = 8) were classified as obese (>30.0).

Moreover, 82.6% (*n* = 19) of patients were postmenopausal, and none of the four premenopausal women resorted to oral contraceptives in the previous 6 months. Only one patient (4.35%) had used antibiotics in the 3 months prior to sample collection.

Concerning the tumour characteristics, one patient (4.3%) had an in situ carcinoma, 34.8% (*n* = 8) of patients had luminal A, and 60.9% (*n* = 14) luminal B BC, 8.7% (*n* = 2) being luminal B-like HER2 positive. Most patients (*n* = 22; 95.7%) had unilateral BC. One patient had bilateral involvement and was tested according to protocol recommendations of germinative testing criteria of the institution for BRCA germinative mutations only, which were negative. One patient (4.3%) had a contralateral metachronous lesion. In respect to the HER2 status assessed by immunohistochemistry (IHC), 34.8% (*n* = 8) were HER2 negative (IHC score= 0), 34.8% (*n* = 8) were HER2 low, 17.4% (*n* = 4) had an IHC score = 1+, and 17.4% (*n* = 4) had an IHC score = 2+ (confirmed HER2 low after SISH). Only two patients were classified as HER2 positive (IHC score= 3).

The TNM staging of the overall population spanned from 0 to IIIC, with stages IA (47.8%; *n* = 11) and IIA (21.7%; *n* = 5) being the most frequent.

Regarding treatment, all patients received endocrine therapy and underwent surgical resection, either tumourectomy (60.9%; *n* = 14) or mastectomy (39.1%; *n* = 9). Additionally, 95.7% (*n* = 22) of women received radiation therapy. A total of 13 patients (56.52%) received chemotherapy: 76.9% (*n* = 10) in the adjuvant and 23.1% (*n* = 3) in neoadjuvant settings. One patient (4.3%) received neoadjuvant anti-HER2 dual blockade and postoperative adjuvant trastuzumab, and one received adjuvant trastuzumab.

Furthermore, other treatments included adjuvant bisphosphonates (21.7%; *n* = 5) and ovarian suppression therapy (4.3%; *n* = 1).

At the time of the last follow-up check-out, two (8.70%) patients had distant disease recurrence.

Further information is summarised in Table 1.

### 3.3. Microbiota Analysis

We first evaluated the richness of the gut microbiota in the two groups. The analysis showed that the community richness was significantly higher in group 1: *p* = 3.28 × 10^–12^ for the Chao1 index and *p* = 1.64 × 10^–12^ for the ACE index. However, the Shannon index, a marker of richness and evenness, was not statistically different between the two groups, *p* = 0.72 (Figure 1).

The β-diversity analysis was used to compare the similarities between samples, including PCoA. The microbiota composition was different between the two groups: a null hypothesis was rejected for PERMANOVA (*p* = 9.99 × 10^–5^) and Anosim (*p* = 0.04) but was not rejected for β-dispersion (*p* = 0.158), using Unifrac weighted distance (Figure 2).

Data analysis using a non-parametric Mann-Whitney *U* test was performed to assess further differential abundance between both groups.

The relative abundances of 14 phyla, 29 classes, 25 orders, 64 families, 116 genera, and 74 species differed significantly between both groups.

The relative distribution of phyla that were found to be significantly different between BC survivors and healthy female controls included Armatimonadetes, Bacteroidetes, Chloroflexi, Fibrobacteres, Firmicutes, Gemmatimonadetes, Nitrospirae, OP11, OP8, Marinimicrobia, Spirochaetes, Saccharibacteria, Verrucomicrobia, and Latescibacteria. Bacteroidetes and Firmicutes were significantly more abundant in group 2 (Figure 3).

Regarding genera relative distribution, Acetobacterium, Acidaminobacter, Acidaminococcus, AF12, Anaerofilum, Arcobacter, Brachymonas, Candidatus Arthromitus, CF231, Desulfurispora, Franconibacter, Frigoribacterium, GW-34, Idiomarina, Kitasatospora, Legionella, Macrococcus, Marinomonas, Microcoleus, Moorella, Natronincola, Oceanicaulis, Paludibacter, Pandoraea, Photorhabdus, Renibacterium, Roseateles, Scardovia, Sharpea, Shigella, Sphaerochaeta, Sulfurimonas, TG5, Thermoanaerobacterium, Tissierella Soehngenia, vadinHB04, and ZA3312c were significantly more abundant in group 1. Akkermansia, Clostridium, Escherichia, Odoribacter, Parabacteroides, Propionibacterium, Streptomyces, and Tannerella were significantly more abundant genera in group 2.

Species with a significantly higher distribution in group 1 were C. acetobutylicum, Desulfotomaculum aeronauticum, Lacrimispora aerotolerans, L. algidixylanolyticum, Halomonas anticariensis, Vibrio atlanticus, Paraclostridium bifermentans, Lactobacillus capillatus, Coprobacillus cateniformis, C. clostridioforme, F. daqui, B. denitrificans, R. depolymerans, S. flexneri, Citrobacter freundii, C. hathewayi, Prevotella intermedia, C. intestinale, Enterobacter kobei, Streptococcus luteciae, Virgibacillus Marismortui, Mitsuokella multacida, P. nigrescens, Bacteroides nordii, Gluconobacter oxydans, P. pallens, Edwardsiella piscicida, Kitasatospora pitsanulaokmensis, Pseudomonas psychrophila, Kosakonia radicincitans, Campylobacter rectus, T. saccharolyticum, C. sartagoforme, Arthrobacter scleromae, S. sonnei, C. sordellii, and P. tannerae.

E. coli, A. muciniphila, C. perfringens, B. stercoris and B. uniformis and Faecalibacterium prausnitzii were significantly more abundant in group 2.

The relative abundance of specific bacterial groups in stools of BC patients and the control group by specific primers is summarised in Table 2.

Furthermore, for group 1, we evaluated the association between β-diversity and clinicopathological factors (age, BMI, HER2 status, menopausal status, and time interval between sample collection and chemotherapy end) using PERMANOVA, Anosim, and β-dispersion tests. In our analysis regarding age (<65, ≥65), BMI (<25, 25–30 and >30), menopausal status, and time from the end of chemotherapy (<2, 2–3, >3–4 and >4 years), a statistically significant *p*-value was not obtained in the division of subgroups. Regarding the HER2 status, our results show a significant *p*-value aggregated to the phylum level (PERMANOVA *p* = 0.009, Anosim *p* = 0.013, and β-dispersion *p* = 0.66).

## 4. Discussion

There is an emerging association between gut dysbiosis and an increased risk of developing malignant diseases, including BC. According to recent data, the human microbiota can play a role in about 15% of cancers worldwide. Furthermore, dysbiosis can affect oncogenesis and tumour progression and influence not only the response, but also the toxicity of antineoplastic therapies [21,22,23].

As mentioned, previous studies have demonstrated a difference in gut microbial profiling between BC patients and healthy individuals. More recent data also support the observation that the microbiota of BC patients differs from healthy controls. Minelli et al. found that, compared to healthy women, those with BC had a different abundance of multiple microbes in the GI tract, including E. coli, Clostridium, Enterobacterium, Lactobacillus, and Bacteroides [24]. The main phyla signatures common to all molecular types identified were Proteobacteria, Firmicutes, Actinobacteria, and Bacteroidetes [10].

Despite some differences, most reports demonstrated a reduced α-diversity in the gut microbiota of women with BC. An abundance of β-glucuronidase-producing bacteria (BGUS) was also evident in BC patients. These types of bacteria, including C. coccoides and C. leptum subspecies, can increase oestrogen reabsorption via the enterohepatic pathway, altering systemic and local oestrogen levels [16]. Terrisse et al. described a higher prevalence of certain bacteria (Eubacterium genera, A. muciniphila, Actinobacteria classes, and Alistipes shahii) associated with the early-stage or node-negative status in BC patients, predicting a slower tumour growth. Meanwhile, the overrepresentation of bacteria species, such as B. uniformis, B. xylanivolvens, and B. intestinalis, was associated with worse outcomes [25].

Bearing this evidence in mind, we analysed the microbiota of BC survivors of a single centre and compared the communities found with those of a healthy control population.

As aforementioned, in our results, the α-diversity was significantly higher in the BC survivors group compared to controls. The microbiota composition (β-diversity) was significantly different between the two groups.

Regarding microbiota diversity, previous data showed contradictory results. As an example, Goedert et al. concluded that postmenopausal BC patients had a statistically significantly altered microbiota composition (β-diversity, *p* = 0.006) and lower α-diversity (*p* = 0.004) versus control patients [17]. More studies that performed microbial profiling analyses of the gut microbiota reported a substantially reduced diversity in BC patients compared to healthy individuals [26,27,28]. In contrast, a metagenomic analysis demonstrated a higher microbial diversity in postmenopausal and no remarkable discrepancy in premenopausal BC patients compared to controls [29]. The high percentage of postmenopausal women in group 1 (82.6%) might partly explain our results.

A decreased gut intestinal bacterial diversity has already been reported in other cancers (e.g., colorectal) and other pathologies, such as inflammatory bowel disease, obesity, metabolic disorders, and extra digestive diseases (e.g., Parkinson’s disease) [30,31,32]. Obesity is associated with increased total fat mass and abdominal fat and decreased lean body mass and affects mostly postmenopausal women, regardless of ageing [33]. We observed a relatively high percentage of overweight/obese women in group 1 (52.2%), with an increased median BMI compared to group 2. Furthermore, two women in group 1 (8.7%) had a history of Crohn’s disease and diabetes mellitus. Moreover, fat percentage seems intrinsically inversely related to A. muciniphila abundance in BC patients. Noteworthy, BC women with a high relative abundance of A. muciniphila have higher levels of Prevotella and Lactobacillus, and lower levels of Clostridium, Campylobacter, and Helicobacter compared to patients with low abundance of the former [34].

In our control cohort, the abundance of A. muciniphila was significantly higher and inversely proportional to BMI compared with the cohort of BC patients. Our results match previous literature, suggesting an inversely related level of A. muciniphila to body fat. Conversely to previous data, the Clostridium genus abundance was significantly higher in the control group, and no significant difference was observed for Prevotella, Lactobacillus, Campylobacter, and Helicobacter.

Additionally, the F/B ratio is widely accepted to have an important influence in maintaining normal intestinal homeostasis and has been used to evaluate gut microbial dysbiosis. Despite the increasing or decreasing of this ratio being a hallmark of dysbiosis, its significance in overweight/obesity is still controversial. Firmicutes and Bacteroidetes represent the Gram-positive and Gram-negative populations, respectively and are the two main phyla involved in the gut metabolism of dietary fibres and polyphenols [35]. Reports with controversial results have associated a higher ratio F/B in obese versus lean subjects [36]. Luu et al. reported a decline in the total bacteria load in overweight and obese patients compared to healthy BMI patients. A significantly lower number of total Firmicutes, F. prausnitzii, Blautia spp., and Eggerthella lenta was observed in patients with overweight/obesity. In the same study, a correlation between gut microbiota and BC clinical stages was performed, with Blautia associated with a higher histoprognostic grade while the C. coccoides cluster, C. leptum cluster, and F. prausnitzii were higher in stage II/III than stage 0/I [37]. These results contrast with other clinical studies [38,39,40].

Chan et al. investigated the microbial population of the nipple aspirate fluid of healthy volunteers and BC survivors using 16S rRNA gene sequencing and reported a lower F/B ratio in BC survivors compared to healthy volunteers [41]. Bobin-Dubigeon et al. characterised the faecal microbiota from early-stage BC patients and healthy controls and observed that the relative abundance of Firmicutes was significantly higher in patients than in healthy controls. In contrast, the Bacteroidetes phylum was significantly more relatively abundant in controls than in patients [42]. In our study, we observed an increased median BMI in the BC survivors’ group. Compared to group 2, our results also showed a significantly decreased ratio F/B in group 1, resulting from a significantly decreased abundance of the phyla Firmicutes and Bacteroidetes. Due to contradictory results in previous reports, the relationship between these main phyla and BMI remains unclear. Our results suggest the existence of other compositional changes at the family, genus, or species level, which might be as relevant as the F/B ratio.

Recent studies found that depressed patients had lower nutrient intakes, poor diet quality, and gut dysbiosis with lower gut microbiota diversity than non-depressed patients. Depressed BC patients had an increased relative abundance of Proteobacteria and a lower Firmicutes abundance than non-depressed patients. Since most bacteria that produce SCFAs belong to the Firmicutes, this may lead to a decrease in SCFAs production, such as butyrate, which may be the physiological basis for intestinal barrier dysfunction and low-level inflammation [43,44]. In our sample, we observed a significantly decreased abundance of the Firmicutes phylum, for which the high percentage (43.5%) of women with a history of depression/anxiety may be a contributing factor. Nevertheless, as these patients’ polimedication regimens also included anti-depressive and anxiolytic medications that could impact the gut microbiota [45], further analysis should be performed to accurately interpret these findings.

Diet quality can shape the gut microbiota composition and functions and affect BC development by impacting the GI microbiota and the microbial and digestive products. Fibres are known to reduce circulating oestrogen levels by altering the gut microbiota and decreasing the deconjugation and reabsorption of oestrogen. Zengul et al. studied the association between dietary fibre and the gut microbiota with a linked BGUS activity in postmenopausal BC patients with newly diagnosed (stage 0–II) in situ or invasive carcinoma. They reported that the total dietary fibre is inversely associated with C. hathewayi, which has been implicated in clinical conditions such as infection and sepsis [46], while soluble fibre is inversely associated with Clostridium. Insoluble fibre was positively associated with B. uniformis, currently used as an indicator of malignancy since it represents an evident characteristic of many cancers due to an expanded glycolytic capability of its strains, related to a higher glucose uptake [47,48]. Our results showed a significantly increased abundance of C. hathewayi in group 1, which may reflect a low dietary fibre intake. A statistically significant decreased abundance of B. uniformis in the same group might shed light on understanding its influence on glycolysis and possibly physiological effects in BC.

Despite nutritional counselling and follow-up by a nutritionist that was provided to patients in group 1, in our study, a detailed questionnaire was not carried out, including specific dietary habits, such as the quantity and quality of fibre ingestion. Future studies including this information will be needed because the influence of diet quality on microbiota composition raises the potential importance of dietary modification in the development and trajectory of BC.

Other oestrogen-independent mechanisms are involved in carcinogenesis that can also be affected by microbiota. Some metabolites derived from fibre fermentation, bile acid (BA), or lipid metabolism interfere with tumour proliferation and differentiation. The BA can induce carcinogenesis via multiple mechanisms, including DNA damage, the activation of the β-catenin signalling pathway, and increased cyclooxygenase-2 (COX-2) activity [11,16,23]. Additionally, it was shown that microbial metabolites could interact with drug pathways, affecting drug metabolism and efficacy through alterations in pharmacokinetics and antitumour toxicity [15]. Specific membrane-bound free fatty acids receptors (FFARs) contribute to the leptin and peptide YY production, secretion of glucose stimulated-insulin, and regulation of inflammatory mediators. FFAR2 and FFAR3 might participate in tumour suppression through propionate and butyrate, influencing cell proliferation and inducing apoptosis [11,49]. SCFAs modulate numerous cancer hallmarks, such as cell proliferation, apoptosis, cell invasion, gene expression, and metabolism in BC [11,18]. A percentage of bacteria with anti-inflammatory characteristics, such as A. muciniphila and F. prausnitzii, can produce SCFAs in a relatively abundant quantity, influencing and determining invasive phenotypes of BC [11,50].

It is well-established that dietary fibre and resistant starch have a potential protective effect against cancer, as both are fermented by the gut microbiota, leading to the production of SCFAs [46]. The three most produced SCFAs include butyrate, propionate, and acetate, each demonstrating anti-inflammatory properties within the host in preclinical studies. Short-chain fatty acids are the most common types of gut metabolites and are primarily produced by the bacterial species E. rectales, C. leptum, F. prausnitzii, and lactate-utilising species E. hallii and Anaerostipes [51]. A decreased level of Roseburia inulinivorans, a bacterium with known anticarcinogenic properties, was reported to be related to a reduction in colonic butyrate and increased inflammation in postmenopausal women, therefore potentially increasing BC risk [4]. The abundance of A. muciniphila, a key player in propionate production, is associated with the richness of the gut microbiota in patients with BC [33]. We observed, in our results, a statistically significant reduction in A. muciniphila and F. prausnitzii abundances in group 1, and no significant difference regarding the abundance of R. inulinivorans, E. rectales, C. leptum, as well as E. hallii and Anaerostipes between both groups, which might suggest a role for SCFAs levels in BC survivorship.

The association between some factors (e.g., diet) and some pathologies (e.g., obesity and depression) that can contribute to gut microbiota dysbiosis and the risk of BC are yet to be clarified, as well as the role of specific taxonomic groups. Despite the small sample, our results suggest an association between BC survivorship and increased gut microbiota diversity, as well as a significantly lower abundance of A. muciniphila, Clostridium genus, and B. Uniformis. However, it is not possible to determine whether it is the consequence or the cause of a favourable disease process.

As mentioned, the gut microbiota metabolises and secretes hormone-like bioactive compounds that modulate BC risk, such as reactivated oestrogens, active phytoestrogens, SCFA, lithocholic acid, and cadaverine [11]. These metabolites are bioactive and act through various pathways that involve the modification of gene expression or modulation of signal transduction in the host [11]. In addition to obesity, immune regulation, the metabolism of other endogenous and exogenous substances, and other factors involved with the Gl microbiota, oestrogen is one of the most important factors in BC development.

Plottel and Blaser defined “estrobolome” as the total sum of bacterial genes in the GI tract capable of metabolising oestrogens [52]. Recently, an atlas of BGUS in human GI tract microbe revealed 3,013 total and 279 unique microbiome-encoded BGUS proteins, clustered into six classes expressed in four bacterial phyla, namely Bacteroidetes, Firmicutes, Verrucomicrobia, and Proteobacteria. Among them, the Bacteroidetes phylum presents the highest abundance and diversity of BGUS enzymes [53]. An estrobolome enriched with enzymes such as BGUS could play a major role in producing free oestrogen, which may increase the risk of hormone-dependent BC in women. In our study, the relative abundances of Firmicutes, Bacteroidetes, and Verrucomicrobia phyla were significantly lower in BC survivors.

Many BGUS-producing bacteria are found in two dominant clusters, namely the C. leptum and the C. coccoides, which belong to the Firmicutes phylum. The Escherichia/Shigella bacterial group, a member of the Proteobacteria phylum, also possesses BGUS enzymes [54]. In particular, in several studies, the abundance of BGUS-producing bacterial species was increased in BC patients, including Clostridium species, compared to healthy controls [27,28,52]. Similarly, the BGUS-producing bacterial species were detected in the nipple aspirate fluid of BC survivors and have been found to increase the time that deconjugated compounds within the host remain in circulation [41]. Zhu et al. reported that, among the 45 species for which the relative abundance differed significantly between postmenopausal healthy controls and BC patients, 38 of them were enriched in postmenopausal BC patients, including E. coli, Klebsiella sp_1_1_55, P. amnii, Enterococcus gallinarum, Actinomyces sp. HPA0247, Shewanella putrefaciens, and Erwinia amylovora. The latter two were shown to have a weak but positive correlation with oestradiol, suggesting a potential involvement in oestrogen metabolism. On the other hand, seven species were more abundant in healthy controls, including R. inulinivorans [29].

In our study, the abundance of the genus Clostridium, and some recognised species with BGUS enzymes, such as C. perfringens, E. coli, and B. uniformis were found to be significantly higher in group 2. Despite the limitation of our small sample, our results may suggest a potential role of BGUS in hormone-dependent BC survivorship.

It is of extreme importance to delineate factors that beneficially or negatively impact the efficient activation of anticancer responses because, as it is known, they are one of the keys to successful treatment outcomes. Interestingly, the microbiota dysbiosis and the differences found between the survivors and the control group could be explained, not by a cancer effect, but by a treatment effect.

The impact of chemotherapeutic agents on gut microbiota has been described, namely a reduction of the diversity, an increased abundance of potentially pathogenic microbes, such as E. coli and Pseudomonas spp., and decreased abundance of Gram-positive bacteria, such as Bifidobacterium and Lactobacillus [55]. Huang et al. described a decrease in 29.6% of total gut microbial content among children who received chemotherapy [56]. Although conflicting results, an increased abundance of Firmicutes and a reduced abundance of Bacteroidetes have been described [55,56]. In our study, 13 (56.5%) women received chemotherapy. A decreased abundance of Firmicutes, E. coli, Bifidobacterium and Lactobacillus was observed in group 1. Despite (18)F-fluorodeoxyglucose, (18)F-FDG positron emission tomography/computed tomography (PET/CT) is not used for diagnosis, staging, and monitoring response to treatment in our clinical practice routinely, we know that physiologic intestinal uptake may reflect subclinical inflammation and differences in the composition of the gut microbiome in BC patients. Tiberio et al. investigated the correlation between inflammatory habits with baseline bowel (18)F-FDG uptake and with a pathological complete response (pCR) to neoadjuvant chemotherapy and concluded that patients’ eating habits affected bowel (18)F-FDG uptake and that colon SUVmean correlated with pCR, identifying a PET scan as a possible instrument for the detection of unhealthy behaviors, such as pro-inflammatory foods and drinks [57]. Some evidence also suggests that nutritional intervention may be an integral part of the multimodal therapeutic approach, as a key factor in determining the efficacy of anti-tumor therapies [58].

Regarding ET, widely used in luminal BC, there is evidence associating letrozole, an aromatase inhibitor, with a shift in gut microbiota and a reduction in diversity and phylogenetic richness [11]. In our study, all women enrolled received endocrine therapy, including 12 (52.2%) with letrozole. As mentioned, in group 1, our results showed an increased α-diversity and a lower relative distribution of Bacteroidetes and Firmicutes phyla and some BGUS, such as C. perfringens, E. coli, and B. uniformis. As mentioned, oestrogen is regarded as a major determinant of BC pathogenesis through both oestrogen receptor-dependent and independent pathways, with gut BGUS having an essential participation in oestrogen reactivation activity. It is apparent that both cancer itself and anticancer therapies, such as ET, interact with gut microbiota bidirectionally. The contribution of ET to BGUS relative lower distribution in group 1 and its role as a potential biomarker in BC pathogenesis and survivorship is not yet clarified. However, once more, it is not possible to determine if our results are a consequence or a possible cause of a favourable response to treatment. Moreover, further studies are needed to investigate the potential role of the gut microbiota in response to the treatment, including chemotherapy, endocrine, and targeted therapies.

The gut microbiota can potentially facilitate or prevent carcinogenesis and may influence an individual’s response to specific cancer therapies. In HER2+ BC patients who received a neoadjuvant treatment (chemotherapy with trastuzumab), those who were non-responsive had lower α-diversity, low levels of Lachnospiraceae, Prevotellaceae, Actinobacteria (Bifidobacteriaceae), Turicibacteriaceae, and Desulfovibrio, and more Bacteroides than patients who achieved a pathological complete response [11,16]. In our study, there are only two patients with the subtype of HER2+ BC, leading to a very small sample to analyse the gut microbiota and the impact of anti-HER2+ therapy.

In our preliminary analysis, we identify as a bias the different sample sized groups. We are currently still recruiting BC survivors for group 1, which will allow us to increase the number in the group 1 population and report our final results. The current analysis is limited by several factors, including a difference in median ages between the two groups. Our control group was around 10 years younger than group 1 and only 17.4 % (*n* = 4) of BC survivors were older than 65. This may have interfered with our results since age is one of the factors causing changes in the intestinal microbiota composition and alterations in gut microbiota were mainly observed in elderly populations (>65 years) [53,59,60].

The different geographical backgrounds of the groups are another limitation of our study. Group 1 is composed of women living in Portugal and was compared to a control group of healthy American females, which might result in different dietary habits that may affect the composition and function of the gut microbial communities [61]. In particular, gut favourable microbiota species can be replaced by toxic metabolites due to a more Westernised diet, characteristic of an American population [62]. Another potentially limiting factor is the use of antibiotics in the previous 3 months prior to sample collection, in the case of one woman of group 1, since its use can lead to the improper development of the host immune system, as well as the depletion of healthy gut microbiota [63].

It is also worth mentioning that we used 16S rRNA gene sequencing, a popular taxonomy profiling choice. However, the resolution accuracy of 16S rRNA gene sequencing to a species level depends wholly on the regions targeted, detecting only part of the gut microbiota community revealed by shotgun sequencing, which allows for the more secure identification of taxonomy at the species and strain levels [64]. Furthermore, the bioinformatics pipeline used was the same for all samples, but regarding the methodology used, the control group samples were processed using the EMP protocols with a set of V4 region publicly available primers and sequenced using Illumina technology. Therefore, it is possible that some value fluctuations in the comparison of the control and the BC survival groups can be explained by these factors.

Another possible limitation of our study is the lack of a complete metabolomics analysis concerning bioactive metabolites that can interact through various pathways involved in gene expression in gene expression modification or the modulation of signal transduction in the host.

## 5. Conclusions

Due to the evolution of medical treatments, BC survivors are a relatively contemporary and growing population. Consequently, factors influencing the onset and progression of the disease have become pressingly important. In addition, these patients comprise a unique group, where both individuals and physicians face the challenges of dealing with multiple long-term side effects of treatment protocols and with ongoing monitoring of disease recurrence. By testing the differences between BC survivors and healthy controls, our study allowed us to observe significant taxonomic disparities in the two groups, which should contribute to an even more active reflection on their causes and consequences.

As in the literature, there is still limited and scattered data regarding microbiota alterations in BC survivors. Additional studies are needed to clarify the involved mechanisms and explore the relationship between microbiota and BC survivorship. Nevertheless, the available evidence seems to reinforce the importance of better exploring this field extensively. Henceforth, it is essential to better comprehend the dysbiotic system’s influence on an individual. Furthermore, microbiota could be a potential predictive or prognostic biomarker for BC survivorship, thus paving the way towards even more targeted treatment protocols with progressively better outcomes and improved quality of life.

## Figures and Tables

**Figure 1 cancers-15-00594-f001:**
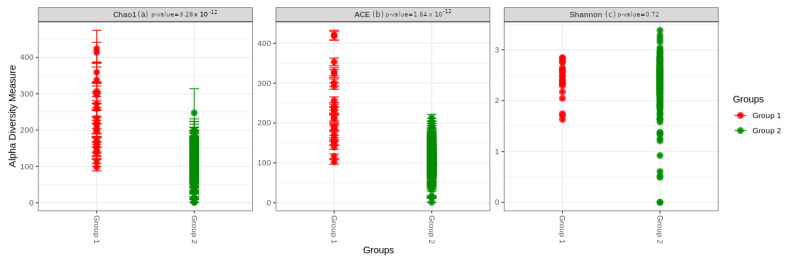
Alpha diversity indices boxplot between group 1 (BC survivors) and group 2 (healthy controls). Comparison based on the (**a**) Chao, (**b**) ACE, and (**c**) Shannon indexes.

**Figure 2 cancers-15-00594-f002:**
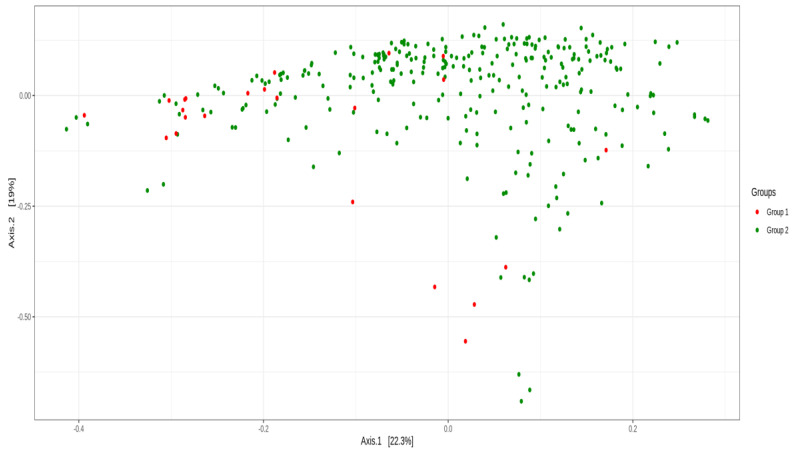
β-diversity analysis of principal coordinate analysis (PCoA) using Unifrac weighted distance between group 1 (BC survivors) and group 2 (healthy controls).

**Figure 3 cancers-15-00594-f003:**
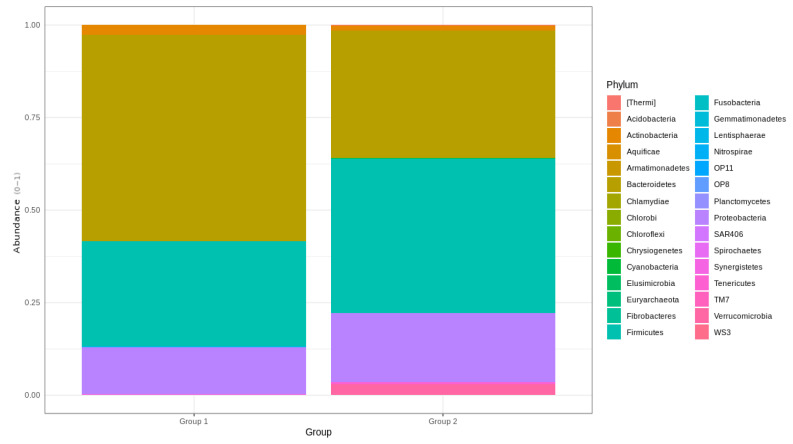
Barplots of top 30 phylum found in the gut microbiota of the BC survivors and control group. Relative frequencies normalized to a 0–1 scale.

**Table 1 cancers-15-00594-t001:** Characteristics of patients at baseline.

Characteristics	Study Group (*n* = 23)
Age—yr	
Median Range	53.00 38–76
Female sex—*n* (%)	23 (100)
Race—*n* (%)	
White Black	21 (91.3) 2 (8.7)
Body Mass Index (Kg/m^2^)—*n* (%)	
Median Range <25 25–30 >30	26.10 21.60–41.60 11 (47.8) 4 (17.4) 8 (34.8)
Menopause—*n* (%)	
Yes No	19 (82.6) 4 (17.4)
Antibiotic use in the prior 3 months—*n* (%)	1 (4.35)
Breast involvement—*n* (%)	
unilateral bilateral	22 (95.7) 1 (4.3)
Tumour type—*n* (%)	
Luminal A Luminal B HER2 negative Luminal B HER2 positive	8 (34.8) 12 (57.2) 2 (8.7)
HER2 status (IHC)—*n* (%)	
0 1+ 2+ 3+	8 (34.8) 8 (34.8) 4 (17.4) 2 (8.7)
Surgical treatment—*n* (%) Tumourectomy Mastectomy Radiation therapy	23 (100) 14 (60.9) 9 (39.1) 22 (95.7)
Endocrine therapy	23 (100)
Chemotherapy Adjuvant Neoadjuvant	13 (56.5) 10 (76.9) 3 (23.1)
Neoadjuvant dual HER2 blockade + adjuvant trastuzumab	1 (4.3)
Adjuvant trastuzumab	1 (4.3)
Ovarian suppression therapy	1 (4.3)
Adjuvant bisphosphonates	5 (21.7)

**Table 2 cancers-15-00594-t002:** Relative abundance of specific bacterial groups in stools of BC survivors and control group by specific primers. The results are expressed as median % (25–75th). *p* value ≤ 0.05, significant difference between the two groups, Mann-Whitney *U* test.

Bacterial Population	Study Group (*n* = 23)	Healthy Controls (*n* = 291)	*p*
Bacteroidetes phylum	22.03 (15.84–34.01)	34.40 (21.95–45.05)	<0.001
Firmicutes phylum	11.23 (8.07–17.33)	41.61 (28.78–54.27)	0.02
Verrucomicrobia phylum	0.006 (0–0.0008)	2.89 (0.049–2.81)	<0.001
Actinobacteria phylum	1.06 (0.04–1.1)	1.42 (0.2–1.3)	0.14
Proteobacteria phylum	5.03 (2.16–6.67)	18.64 (1.80–27.31)	0.18
*Clostridium* genus	0.13 (0–0.12)	0.6 (0–0.075)	0.02
*Prevotella* genus	7.21 (0.21–17.77)	5.66 (0.07–3.91)	0.07
*Shigella* genus	0.19 (0–0.13)	0.002 (0–0)	<0.001
*Lactobacillus* genus	0.004 (0–0.001)	0.11 (0–0.01)	0.92
*Bifidobacterium* genus	0.36 (0.0003–0.17)	0.64 (0.01–0.6)	<0.001
*Roseburia inulinivorans*	1.28 (0.41–1.74)	1.53 (0.21–2.34)	0.21
*Akkermansia muciniphila*	0.006 (0–0.0006)	2.89 (0.49–2.81)	<0.001
*Clostridium perfringens*	0.2 (0.001–0.288)	0.34 (0–0.48)	<0.001
*Escherichia coli*	0.01 (0–0.05)	0.03 (0–0)	<0.001
*Bacteroides uniformis*	2.31 (0.27–3.68)	6.73 (1.14–9.5)	<0.001
*Clostridium hathewayi*	0.004 (0–0)	0	<0.001
*Faecalibacterium prausnitzii*	7.36 (1.23–15.75)	27 (16.31–37.45)	0.002

Firmicutes/Bacteroidetes ratio was calculated for both groups and was significantly lower in group 1 (0.5096) than in group 2 (1.2098), *p* < 0.001.

## Data Availability

Data is unavailable due to privacy or ethical restrictions.

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
