# Peer review of "Breast Cancer Survivors and Healthy Women: Could Gut Microbiota Make a Difference?—“BiotaCancerSurvivors”: A Case-Control Study"

_cancers, 2023, doi:10.3390/cancers15030594_

Round 1

Reviewer 1 Report

Explain how the sample size calculation was conducted.

Why the authors did not consider steroid or antibiotic use in the month prior to fecal sample collection as possible exclusion criteria? One might nonetheless consider reporting the data on steroid and antibiotic that the authors claim to have collected.

I recommend excluding stage 0 from the cases, as it is not an infiltrating breast carcinoma

In Table 2 I would recommend including the demographic characteristics of the controls and performing a sensitivity analysis

In the table, I would also simplify the tumor subtypes into luminalA, luminalB HER2 + or -.

In the discussion the authors report the association between dysbiosis and risk of developing breast cancer but also to probability of different response to therapy. The authors should briefly expand this aspect and they could use this evidence:

The Relationship among Bowel [18]F-FDG PET Uptake, Pathological Complete Response, and Eating Habits in Breast Cancer Patients Undergoing Neoadjuvant Chemotherapy

Diet and breast cancer: Understanding risks and benefits. Physiologic intestinal 18F-FDG uptake is associated with alteration of gut microbiota and proinflammatory cytokine levels in breast cancer.

Nutrition and Breast Cancer: A Literature Review on Prevention; Treatment and Recurrence.

I suggest matching between cases and controls by balancing for clinical features or otherwise discussing whether or not it can be demonstrated in the present study that the observed differences are indeed related only to the prior diagnosis of breast cancer and not to different clinical features of the two groups.

Finally, I suggest more effectively summarizing the various points (all relevant) of the Discussion section, eventually also with the support of tables.

Reviewer 2 Report

Thank you very much for giving me the opportunity to review the paper entitled “Breast Cancer Survivors and Healthy Women: Could Gut Microbiota Make A Difference? - “Biotacancersurvivors”: A Case- Control Study” by Caleça T et al. Although this is a fair and coherent experimental study, some issues and comments need to be addressed:

MAJOR:

1. The only major comment related to the proposed draft paper, refers to the heterogeneity of the sample size between group 1 and 2. If similar sample sized groups would have been used, we could extrapolate that the presented results would have been significantly different from the analytic statistics’ point  of view. If the authors could address this as being not a limitation, but, as a potential statistical bias of the study, in the Discussion section, the paper would gain relevance and power. Otherwise, we could end up in “comparing few pears to many apples”-situation. Or, the authors could at least, argument why their study groups (group 1) has such a “gap” in sample size number, or consider recruiting more BC patients form other Portuguese cancer centers. They have superbly managed to explain the discrepancies between the 2 groups regarding their provenience (US and Portugal); it would be a pity not to address the sample-size discrepancies issue, as a potential statistical bias.

MINOR

1. line 207 – please specify why other non-BRCA mutations were not tested in the BC group

2. line 219 – “posterior” is not an adequate term for the context; please consider revising to either “postoperative” or “subsequent”

3. line 223 – consider revising “last contact” to “last follow-up check-out/outpatient admittance etc” or similar

4. Lines 228-229 – consider expressing p values using real 0.value number and not the mathematical powered value generated by SPSS

Otherwise, my sincere congratulations for the proposed paper

Round 2

Reviewer 1 Report

 -